# Older adults experiences of transitioning to housing following homelessness from a perspective of ontological security: A secondary analysis

Rebecca Goldszmidt[1]*, Shu-Ping Chen[2], Rebecca Gewurtz[3], Carri Hand[4], Carrie Anne Marshall[1]

1 Faculty of Health Sciences, Social Justice in Mental Health Research Lab, Western University, London, Ontario, Canada, 2 Faculty of Rehabilitative Medicine, Department of Occupational Therapy, University of Alberta, Edmonton, Alberta, Canada, 3 Faculty of Health Sciences, School of Rehabilitation Science, McMaster University, Hamilton, Ontario, Canada, 4 Faculty of Health Sciences, School of Occupational Therapy, Western University, London, Ontario, Canada

* Rgoldszm@uwo.ca

## Abstract

Older adults are increasingly becoming unhoused. Once homeless, older adults are forced to navigate the transition to housing, an area which requires further research. We conducted a secondary analysis of qualitative research data using the theory of ontological security to explore the question: How do older adults experience trauma across the transition to housing following homelessness? During the analysis, we created a central essence 'When your entire world is upended you have to learn to live in a new reality', with three underlying themes: 1) My life got "pulled out from under me" and I am struggling to regain even footing; 2) I see reality clearly, "the system is broken"; and 3) The importance of social connection in rebuilding ontological security. Findings indicate the transition to housing presents an opportunity for older adults to rebuild ontological security, however only when they have access to housing that is good quality, deeply affordable, and accessible.

Older adults are experiencing homelessness at an increasing rate internationally due to a failure of societal structures including the rising cost of living and changing labor markets [1–6]. A fundamental facet of the experience of homelessness is trauma [7–9] and this is especially true for older adults [10–12]. Trauma has also been identified as a key influence on how one experiences transitioning to housing following homelessness, although this research has not focused on older adults. One way trauma can influence this transition is through disrupting ontological security, the sense that the world and the way that it is constructed in one's personal life can be depended upon to remain constant [29,30]. Studies focused on the transition out

**Data availability statement:** Our data contain potentially sensitive information that may be a risk to the confidentiality of participants. For these reasons, we are unable to upload to a repository. Instead, we have indicated that researchers may contact the ethics office at Western University to obtain the data set. The email address for the ethics office is 'ethics@uwo.ca'.

**Funding:** This work was supported by an Ontario Graduate Scholarship (to RG). The funders had no role in study design, data collection and analysis, decision to publish, or preparation of the manuscript.

**Competing interests:** The authors have declared that no competing interests exist.

of homelessness, which includes finding housing, moving into housing, and settling into housing, indicate that finding good quality housing is a challenge and that once housed, individuals may struggle with continued stigma and discrimination, lack of adequate access to nutritious food, and trauma memories [13–15]. It is also important to consider that at least 10–20% of individuals who transition to housing subsequently return to living unhoused [16], and that the transition process is therefore complicated and can involve multiple attempts at finding and maintaining housing. As the number of older adults trying to find housing following homelessness is increasing [6], it is important to better understand how trauma is experienced by older adults across the transition to housing following homelessness so that research, practice and policy can best support this population.

### Trauma and ontological security

From a systems-level perspective, trauma can be understood as the result of structural and interpersonal violence [17]. When we recognize that people who live in unpredictable and unsafe environments need to learn to protect themselves, behaviours that are often considered problematic can be seen as a result of structural violence, rather than as a result of individual pathology or immorality [17–20]. A social justice perspective of trauma draws attention to the connection between trauma and structural violence. Understanding trauma through a lens of structural violence highlights the patterned nature of violence and calls out structures and systems that are experienced by some as traumatic. However, while violence, both structural and interpersonal, that leads to trauma is an issue of social justice, the experience of trauma is often very personal.

Trauma is a deeply personal experience, and one that has been shown to affect people neurobiologically, mentally, emotionally, and spiritually [21–25]. From an individual perspective, trauma can be defined as resulting from "an event, series of events, or set of circumstances that is experienced by an individual as physically or emotionally harmful or life threatening and that has lasting adverse effects on the individual's functioning and physical, mental, social, emotional, or spiritual well-being" (p. 7) [26]. When people experience trauma, their sense of the world and the beliefs they previously depended on may be disrupted, causing a loss of ontological security [27,28]. Anthony Giddens, who wrote extensively on ontological security, defined it as "the confidence that most human beings have in the continuity of their self-identity and in the constancy of the surrounding social and material environments of action." [29]. Ontological security is an important component of mental well-being, as without it, most individuals cannot predict the environmental circumstances in which they are embedded, leaving them with experiences of existential anxiety, feelings of hopelessness and difficulty navigating day-to-day life [29–32].

The theory of ontological security has previously been applied to experiences of trauma, and specifically trauma related to both aging and housing insecurity, to gain insight into how these experiences impact well-being. For instance, in a qualitative study exploring the experience of having a stroke, Alaszewski et al., [33] found that the unexpected nature of having a stroke, coupled with a changed ability to care for

oneself, disrupted older adults' sense of their own health and shook their sense of self, leading to poor mental well-being. Ontological security theory was thus used to bring attention to how having a stroke could be psychologically traumatic, as well as physically traumatic, by disrupting a person's belief in their continued good health. Ontological security theory has also been used in other qualitative studies to describe how being let down by systems that were supposed to provide support can be experienced as traumatic [28,31,33,34]. For example, in the wake of hurricanes Irma and Maria, Aranda et al., [22] found that Puerto Rican participants experienced a loss of trust when their government failed to support them in a timely manner, leading to a sense of societal betrayal and a lack of continued trust in social institutions. Similarly, focusing on the aftermath of hurricane Katrina, Hawkins et al., [28] highlighted how the theory of ontological security could lead to novel suggestions for interventions by bringing attention to the psychological harm that loss of ways of life and routines can have on a person. From the perspective of ontological security, the need for interventions focusing on reconstructing identity, creating community, and creating a new sense of safety are essential [34]. Finally, studies have used the theory of ontological security to explore how the experience of living in insecure housing can be traumatic. Specifically, studies found that living in constant uncertainty led to a lack of trust in the future and a sense that reality was impermanent, which led to poor mental well-being [32,35,36]. Taken together, these studies indicate that trauma can disrupt ontological security. Further, when coupled with a structural understanding of trauma, it can be understood that societal systems including racism and classism put certain groups of people at far greater risk of experiencing certain traumas, making this an issue of social justice [17].

## Ontological security and homelessness

The Canadian Observatory on Homelessness defines homelessness as: "the situation of an individual, family or community without stable, safe, permanent, appropriate housing, or the immediate prospect, means and ability of acquiring it" (p. 1) [37]. Within this definition, it is important to recognize that being homeless is not only about lacking physical housing. Hsieh [38] indicates that "being homeless is deeply emotional and highly personal. It involves existential anxiety about identities, relationships, survivorship, and future outlooks" [38]. As Hsieh [38] contends, being homeless is closely tied to a lack of ontological security [29], in which individuals are forced to live in a state of uncertainty that tests their understanding of themselves and the world.

Other researchers have also used Giddens' [29,30] theory of ontological security to understand the experience of living without housing security, ultimately demonstrating that merely having a temporary roof is not sufficient for establishing ontological security. For example, through interviews with people who had experience living in insecure housing, Watt [35], found that the continued loss of housing or possessions can lead to the feeling that reality is impermanent and that one could lose everything at any moment. With a similar population, Rosenberg et al., [39], found that the constant fear of losing housing meant individuals struggled to plan for their future and feel secure and safe, making it challenging to establish ontological security. Researching the experience of previously unhoused youth, Skobba [40] found that key conditions preventing youth from experiencing ontological security were a lack of control and a lack of constancy. While finding housing may support older adults in gaining a roof over their heads, studies with other populations who have experienced housing insecurity indicate this alone is insufficient for rebuilding a sense of ontological security [32,40]. It is therefore important to consider the experience of older adults from a perspective of ontological security so that research, practice, and policy can best support this population during the transition to housing following homelessness.

## The current study

Older adults are increasingly experiencing homelessness [2,5,6,41]. However, research focused on the transition to housing following homelessness has insufficiently attended to the experiences of older adults [42,43], creating a key gap in understanding and leading to inadequate policies and practices geared towards supporting older adults transitioning

to housing following homelessness. We aimed to address this gap. Informed by the theory of ontological security, we explored experiences of trauma among older adults transitioning to housing following homelessness by conducting a secondary analysis of existing research data. The research question that guided this analysis was: How do older adults experience trauma across the transition to housing following homelessness?

## Methodology

### Ethics statement

The primary study from which this secondary analysis was conducted received ethics approval from both Western University's Human Research Ethics Board (REB# 114922) and Queens University's General Research Ethics Board (REB# 6036610). Written formal consent was obtained from all participants for participation in the study and use of their data in future studies. To answer our research question, we conducted a secondary analysis of qualitative data [44], from interviews by Marshall et al., [45]. A secondary analysis is a method in which interviews from a primary study are used as the data for a new study with a different, but similar, research question. In this case, the primary study focused on what people needed to thrive following homelessness [45]. For the current study, only a subsection of the interviews, those with older adults, were used. Secondary analyses offer benefits, including reducing participant burden, saving costs, and making use of the information the participants shared to promote their voices [44]. However, there are also some drawbacks that should be considered. Beck [44] warns against failing to recognize the context in which the primary data was collected and misinterpreting the findings in a changed political and social climate. Fortunately, these challenges could be navigated in this particular study as the last author led the primary study and acted as the senior researcher on this secondary analysis and the first author was a research assistant on the original study and helped with the interviews. As such, we were familiar with the context in which the primary data was collected, thus mitigating any concerns [44].

### Paradigmatic framework

For the current analysis, we utilized an interpretivist stance [46], which was also used in the primary study. Congruent with an interpretivist stance, we took a relativist ontological position, which holds that reality is constructed in the minds of individuals and through dialogue and experiences [46]. Further, consistent with an interpretivist stance, we held an epistemological position of subjectivity and transactionalism [46,47]. In other words, rather than attempting to find some objective reality, we sought to interpret, through the lens of ontological security, how participants experienced trauma across the transition to housing following homelessness. As such, our analysis is a particular understanding that brings together the varied perspectives of the participants as shared through semi-structured interviews, our own beliefs, and the theory of ontological security.

### Positionality

As researchers we brought varying perspectives to the current study. We used these varying perspectives to reflect on our understanding of homelessness and aging and consider aspects of the participants' experiences we may not have recognized as individuals, such as the impact of societal discourses around ageing. There are also commonalities that led to us, as researchers, having a similar perspective. We are all women, and not considered older adults, which influenced our understanding of the participants experiences. Further, we purposefully took a perspective of social justice, which underscored the entire study and informed our belief that homelessness is an issue of social justice that needs to be eliminated by providing everyone with the capabilities they need to live a meaningful life [48]. To be reflexive about the ways our experiences influenced the current study, the primary author kept a reflexivity journal and as researchers we considered when we were unconsciously bringing in ageist or ableist views and when we were purposefully taking a perspective of social justice.

## Recruitment

The primary study from which this secondary analysis was conducted received ethics approval from both Western (REB# 114922) and Queens Universities (REB# 6036610). During the consent process, participants were explicitly asked to consent to the use of their data in future research. Participants were recruited from two mid-sized cities (i.e., London, Ontario and Kingston, Ontario) and were recruited from places frequented by individuals experiencing housing precarity, through working with gatekeepers such as leaders and case managers in health and social care organizations and through attending drop-in times at local organizations. Further details can be found in the primary study [45].

## Inclusion and exclusion criteria

The original study recruited participants who were over the age of 16, gave consent, and had experiences of being unhoused for at least one month within the past three years as well as an acknowledged substance use disorder or mental illness [45]. For the current study, we excluded all participants who were under age 50 at the time of the interview. We utilized age 50 as a cut-off point following the lead of Grenier and colleagues [49], who, in a literature review on homelessness and aging, suggested that, due to increased physical aging, homelessness researchers should define older adults as those aged 50 and older. This convention has also been used by multiple researchers including Canham et al., [50], Brown et al., [1,51], and Crane & Warnes [2].

## Data collection

Participants in the primary study were recruited between June 26th, 2020, and December 15th, 2020. Participants were asked to provide written consent, both for the primary study and any future use of their data in research and choose a pseudonym to ensure confidentiality. Participants then provided information on their demographic and health characteristics including: age; gender; sexual orientation; race and ethnicity; income; self-reported mental health conditions; and housing status. Participants were also asked about their substance use using two measures; the Alcohol Use Disorders Identification Test (AUDIT-10) [52] and the Drug Abuse Screening Test (DAST) [53]. A score of 8 or higher on the AUDIT indicates that a person is engaged in hazardous alcohol use. Scores on the DAST range between 0–10 with higher scores indicating greater substance use. Following the demographic questions, semi-structured interviews were conducted.

In the current study interviews (n = 15) lasted between 12–62 minutes (m = 39; sd = 12.8) and focused on the topic of transitioning out of homelessness, addressing services, housing, community integration, and mental well-being and substance use. As participants were both housed and unhoused at the time of the study, we were able to gain insight from across the trajectory of unhoused to housed, including the preparation, finding housing, and adjusting to housing, and the cycle of subsequently losing and then finding new housing. A sample of the interview guide used in the primary study is provided in Appendix A in S1 Text. More detailed information about the process of recruitment and data collection can be found in the primary study [45].

## Secondary analysis

We conducted an abductive reflexive thematic analysis following the guidelines of Braun and Clarke [54] and using the theory of ontological security described by Giddens [29,30]. The first author (RG) engaged in coding using the program Dedoose [55]. Coding involved first listening to the audio interviews and reading and re-reading the transcripts to gain familiarity, an important element of rigor [56]. Secondly, initial codes were created by considering sections of the transcripts through the lens of the research question. During this process, collaboration between the first author (RG) and the last author (CM) led to the utilization of the theory of ontological security [29,30] as a perspective through which to interpret the analysis. Consistent with an abductive approach [57], multiple theories were considered and discarded during this process until the theory of ontological security was decided on due to its ability to help understand the participants'

experiences. The theory of ontological security [29,30] was subsequently used in the third stage as codes were grouped together into subthemes and then larger themes. Specifically, the theory of ontological security influenced the ways we choose to name and consider themes and how we interpreted different experiences. For instance, we interpreted participants discussion around feeling stressed, uncertain, and depressed in new housing due to their routines and ways of interacting with the world being disrupted as an indication that participants were experiencing a lack of ontological security, which was impacting their mental wellbeing. Fourthly, following the guidance of Braun and Clarke [54], a central essence that connected the themes was created. The final stage of the analysis was the writing, where we further considered each theme in relation to ontological security and incorporated participant quotes to center the participants' experiences.

## Findings

### Sample characteristics

Fifteen participants met the criteria for the secondary analysis. The mean age was m = 58.53; sd = 5.9 years, range = 50–68. The gender of the participants was as follows: n = 11 (73.3%) men; n = 4 (26.7%) women; n = 0 (0%) other genders. The race of the participants was: n = 13 (86%) White; n = 1 (7%) Hispanic; n = 1 (7%) Metis. When participants were asked if they identified as 2SLGBTQIA+, n = 14 (93%) participants responded 'No' and n = 1 (7%) responded 'Yes' and identified as bisexual. The housing status of the participants was: n = 10 (66.7%) unhoused; n = 5 (33.3%) housed. Of those living unhoused at the time of the interview, participants reported staying: n = 9 (60%) in shelters; n = 4 (27%) outdoors; n = 2 (13%) couch surfing; n = 1 (7%) "here and there"; n = 1 (7%) at warming/drop-in centers. Of those that were housed, n = 2 (13%) lived in cluster site permanent supportive housing; n = 2 (13%) lived in market rent units; and n = 1(7%) lived in scatter site permanent supportive housing. For income, n = 10 (67%) received disability-related government income support (ODSP); n = 3 (20%) received Ontario Works (OW); n = 3 (20%) had a Canadian pension plan; n = 2 (13%) received old age security. Other ways of generating income identified by participants included being self-employed (n = 2, 13%); panhandling (n = 1, 7%); bottle collecting (n = 1, 7%); babysitting (n = 1, 7%); and making and selling artwork (n = 1, 7%).

Participants reported a mean of 3.3 (range = 1–6; sd = 1.7) mental health conditions including: n = 12 (80%) mood; n = 11 (73%) anxiety; n = 11 (73%) stress and trauma related; n = 6 (40%) psychotic; n = 5 (33%) personality; and n = 5 (33%) obsessive compulsive. Participants reported a median score of 1 on the DAST [53] (interquartile range (IQR) = 5), indicating low problematic use of drugs, and a median of 3 on the AUDIT [52] (IQR = 7), indicating low problematic use of alcohol. Further information on the characteristics of participants is presented in Table 1 below.

### Qualitative findings

**Essence: When your entire world is upended you have to learn to live in a new reality.** Going from housed to unhoused means learning to live in a new reality and this is true throughout and following the transition to housing. We generated three themes that express this essence: 1) my life got "pulled out from under me" and I am struggling to regain even footing; 2) I see reality clearly, "the system is broken"; and 3) The importance of social connection in rebuilding ontological security. Throughout the analysis, the theoretical framework of ontological security [29,30] was used to frame the analysis and explain how for older adults with experiences of trauma, major transitions, including the loss of housing and the transition to housing following homelessness were experienced as disrupting one's sense of ontological security that necessitated learning to live in a new reality. Within the findings, participants are identified by self-chosen pseudonyms to provide anonymity.

**Theme 1: My life got "pulled out from under me" and I am struggling to regain even footing.** For many participants, life altering events, including experiences of childhood trauma, age-related health changes, and loss of housing severely altered the way they viewed the world, themselves, and their place in the world. This led to the feeling

**Table 1. Demographic characteristics of participants.**

| Demographic Characteristics (n = 15) | n (%) |
|---|---|
| *Gender* | |
| Men | 11 (73) |
| Women | 4 (27) |
| Other | 0 (0) |
| *Age* | |
| | $M = 58.5$ |
| | $Range = 50–68$ |
| | $sd = 5.9$ |
| *2SLGBTQIA+* | 1(7) |
| *Race/Ethnicity* | |
| White | 13 (86) |
| Hispanic | 1 (7) |
| Metis | 1 (7) |
| *Age at first time Homeless* | |
| < 50 | 5 (33) |
| > 50 | 4 (27) |
| Missing | 6 (40) |
| *Housing status at time of study* | |
| Living without stable housing | 10 (67) |
| Shelters | 9 (60)[a] |
| Street/outdoors | 4 (27)[a] |
| Couch surfing | 2 (13)[a] |
| Warming/drop in centers | 1 (7)[a] |
| Housed after housing instability | 5 (33) |
| Permanent supportive housing | 3 (20) |
| Market rent unit | 2 (13) |
| *Current source of income* | |
| ODSP | 10 (67)[a] |
| OW | 3 (20)[a] |
| Canada pension plan | 3 (20)[a] |
| Old age security | 2 (13)[a] |
| Self-employment | 2 (13)[a] |
| Other | 5 (33)[a] |
| *Mental health conditions* | |
| Anxiety | 11 (73)[a] |
| Mood difficulties | 12 (80)[a] |
| Psychosis | 6 (40)[a] |
| Personality problems | 5 (33)[a] |
| Stress and trauma related problems | 11 (73)[a] |
| Obsessive-compulsive | 5 (33)[a] |
| *Drugs used* | |
| Speed/cocaine/crack/crystal meth | 4 (27)[a] |
| Cannabis | 4 (27)[a] |
| Off-label opioids | 1 (7)[a] |
| *Gib* | 1 (7)[a] |
| *Substance Use* | |

*(Continued)*

**Table 1.** (Continued)

| Demographic Characteristics (n=15) | n (%) |
|---|---|
| Alcohol use (AUDIT) | 0-14 (Mdn=3; IQR=7) |
| Non-hazardous use (<8) | 13 (87) |
| Hazardous use (≥8) | 2 (13) |
| Drug Use (DAST) | 0-8 (Mdn=1; IQR=5) |
| No problem (0) | 7 (47) |
| Low level (1-2) | 2 (13) |
| Moderate level (34-5) | 3 (20) |
| Substantial level (67-8) | 3 (20) |
| Severe level (9-10) | 0(0) |

ᵃpercentages do not add up to 100 because multiple options were possible.

AUDIT=Alcohol Use Disorders Identification Test; DAST=Drug Abuse Screening Test; Mdn=median, IQR=interquartile range.

that their life had been "pulled out from under…" them [Matt], or what Giddens would describe as a loss of ontological security [29,30]. In the wake of such a loss, older adults discussed struggling to learn how to live in a different world and the challenge of finding new housing.

For participants who became homeless later in life, the loss of housing occurred as part of a series of life-altering events, with Matt describing the loss of housing and living unhoused as "…three years of absolutely nothing but devastating". For Matt, these three years included the loss of his parents, his brother selling off the family home which left him homeless, and a slow and challenging search for housing. At the time of the study, Matt was trying to find housing, but was struggling with this process, as most housing options were inaccessible, leaving him in a situation of continued uncertainty. For another participant, Doc, the loss of ontological security occurred when he divorced his wife, leading to living in a series of temporary places with little security, exemplified by his story of how one day his landlord "…dumped me downtown and kept my stuff" [Doc]. Following losing his permanent housing, Doc moved from one bad place to another, trying to regain even footing.

The experience of physical aging while unhoused and searching for housing also seemed to result in uncertainty and mental distress and impacted participants' sense of ontological security. For example, Runnr described how, after experiencing multiple heart attacks, he was constantly preoccupied with ensuring that his cell phone was fully charged because "If I don't have a phone and I have another heart attack…I'm afraid" [Runnr]. Such a preoccupation inhibited his ability to find housing, as he spent all his time and mental energy worrying about his health, stating "I don't get any help I just live day by day". While Lola took a different approach emotionally, stating, "…heart surgery. It's really given me a totally different outlook on life", her focus on living in the moment also impacted her ability to find housing, as she was focused on enjoying life day-to-day, rather than considering the future. Both Runnr's and Lola's approaches appeared to align with Gidden's [29,30] theory on how people cope with a loss of ontological security. Runnr exhibited pragmatic acceptance, and Lola appeared to use sustained optimism, which similar to findings from Power [32] supported their mental wellbeing at the cost of them engaging with the steps to find housing. Further, likely due to her age, Lola was a good target for theft and described how her phone had been stollen at least three times while living in a shelter, and as she no longer had a laptop, it was difficult to search for housing. Age-related disabilities also changed how older adults interacted with the world. Bambi described a traumatic fall down the stairs which resulted in PTSD and made her housing inaccessible. Due to these two factors Bambi left her housing

"…just to be able to do what I want, go to bed when I want, get up when I want. You know, leave, and go to the store or something when I want. Because I can't when I'm at my son's, I just can't do it. Because he has to be there to help me down the stairs"

Bambi discussed how her ideal housing was long-term care, however, as the waitlist was very long, she was also looking for housing geared towards older adults, with no stairs, and support with cooking and cleaning. At the time of the study Bambi was struggling to find such housing, as most housing she was shown had stairs, and discussed how this was demoralizing, stating "Sometimes I do feel not wanted. And I think that's because of not being able to find a place", which meant she was stuck in the shelter. For Bambi, Lola and Runnr, it was not living unhoused that disrupted their sense of ontological security but the combination of health challenges and not being able to find housing that met their needs.

For a subgroup of participants who experienced homelessness long before the age of 50, their sense of ontological security was disrupted through early and lifetime experiences of trauma. Giddens [29,30] describes ontological security as developing during childhood through secure relationships with caregivers and the world, with a loss of ontological security leading to experiences of existential anxiety and not feeling safe in the world. One participant who had not had stable housing in over 15 years described how "a lot of us have lost the actual feeling that I think, that people really do actually care" [Husky]. Experiencing continued trauma with no reprieve took a toll, with one participant stating "…it interrupts my eating, interrupts my sleeping" [Pekoe]. Continued experiences of trauma could lead to distrust and self-protective aggressive behavior that found some participants banned from certain locations. Such was the case for Smiley who described how he had once accepted housing only to find out he was not allowed there, stating, "when I got there to pick up my key um the nice lady said oh, you're not allowed here.". Smiley was not entirely sure why he had been banned but postulated it was due to instances where he engaged in aggressive behaviour to protect someone, which seemed to occur frequently as Smiley did not trust the system to protect other people living unhoused. What unified these experiences was that living in unsafe situations across one's life had disrupted older adult's sense of safety in the world, leading to a lack of trust in people which made it challenging to find housing and stay housed.

**Theme 2: I see reality clearly, "the system is broken".** Participants emphasized how realizing that society would not step in to save them from their current situation of living unhoused and in poverty disrupted their sense of trust in government systems, which indicated a lack of ontological security. In the past, they had the sense that their needs would be met by existing structures, but the failure of those structures shocked them, and left them feeling abandoned: "We're in a hole and we're gonna stay in a hole until we die" [Donny]. Within a society in which some lived in excess, the structural violence of being forced to experience homelessness led to a sense of societal betrayal and injustice exemplified by Pekoe's statement that "the system is broken". Experiences that illuminated this harsh reality and made participants feel insecure were the limited and derelict housing options, the lack of control many participants felt over their own lives and being treated in undignified and stigmatizing ways by those that represent the system. Conversely, being treated well by service providers helped participants maintain or gain back trust in the system.

The limited and derelict housing options were experienced by participants as a societal betrayal in which they were forced to choose between the "…lesser of two evils" [Pekoe]. Derelict housing is a common issue following homelessness that negatively effects individuals mental and physical wellbeing [58]. Poor quality housing includes the physical attributes such as leaks, peeling paint, inadequate plumbing, mold, bedbugs, and more, the social factors such as crime, lack of public transport, and living with unsafe roommates, to name a few [59]. For older adults, poor quality housing also includes inaccessibility due to stairs or distance from necessities. Pekoe described this well stating, "when it comes time to go grocery shopping or to do the laundry or to have to carry anything it's very difficult for me to do and I'll often put those things on the wayside and actually not even go get food and stuff like that because I dread the fact of having to carry anything back." While Pekoe had become resigned to this reality, for new unhoused individuals this reality was challenging for participants to accept, and some initially turned down physically poor-quality housing. For instance, Mojo and Matt both described how when they were offered derelict housing, they turned down these offers: "I don't want that cockroach, bedbug-infested place man, that's not me" [Mojo]; "you want me to give this guy rent money, loco… I don't got a lot of money but take me somewhere where it might be a pleasure to live" [Matt]. Other participants made different choices, accepting poor quality housing to at least feel they had their own place. Participants also described choosing

between different aspects of poor quality such as housing that contained bedbugs rather than unsafe roommates. Either way, being shown poor-quality housing by service providers was described as insulting and over time became "character killing" [Matt], devastating participants' sense of self-worth. Some participants seemed to recognize that this was not the fault of service providers, but rather a system-level problem in which society did not support them, a phenomenon that Wathen and Varcoe [17] refer to as structural violence.

Stigma and discrimination towards unhoused individuals is well documented and especially likely to occur in settings where health care providers do not understand the social context of living unhoused [60]. Stigma and discrimination were described by participants who discussed how they felt they were at the mercy of others' whims, specifically those in positions of power, leaving them with little security. For instance, Doc, who wanted therapy stated, "I've had my prescriptions caught up and that's it and I haven't had anything, anything remotely like therapy for a while". Doc wanted therapy, however only with an older male who might understand his experiences, which was not offered. Casey Jones also described feeling at the mercy of a psychiatrist, saying "and then he wanted to know why I had the eating disorder, and I had to tell him really graphic stuff about my rape and he just, it was not a good scene". This lack of control challenged participants' sense of ontological security, leading to feeling powerless and helpless within a broken system.

While some participants expressed feeling helpless, others discussed the ways they fought back, engaging in what Giddens would call radical engagement [29,30]. Cheech, a Metis man who wanted to survive off the land, discussed how he refused to give up important belongings such as his canoe to receive government support, saying "yeah, well the government says you don't need belongings, you're not supposed to have anything when you're homeless.". For Cheech, his canoe was important to the way he lived his life and being asked to give it up was described as offensive and a continued example of how social structures were designed to constrain and prohibit rather than support. Other participants also described how the loss of belongings in the shelter impacted their feelings of security and safety within these spaces. Such findings were not unique to this study, as across studies older adults have described feeling controlled and unsafe in shelters [61]. Another participant, Donny, demonstrated radical engagement through supporting his community even when it broke the laws, stating, "I go there no matter what, I don't care what anybody says. I'm going in there and I'm going to do what I can to help that person". Neither approach helped with finding housing, but radical engagement did support individuals in staying true to themselves and gaining some element of control in their lives.

Participants described how the experience of aging and going through homelessness was one in which the general public and people in positions of power degraded them, ignored them, and generally made them feel invisible and worthless. Over time, this broke down their sense of ontological security through challenging their sense of identity, self-worth, and ability to trust. Pekoe, discussing trying to get support for PTSD stated, "I scream it out loud…but you might as well be talking to that wall over there", and indicated that the constant dismissal of his pain led to feeling "…unwanted, [and] uncared for." [Pekoe]. Donney, describing his experiences of stigma and how it negatively affected his sense of self, stated, "It feels like everybody's picking on you, you know you're, you're, you're just nothing" [Donny]. Participants also described how hurtful it was that people within society turned their back on them when they were clearly in need. Smiley described how this had happened to him outside a hospital stating "Um, they actually let me go with no shoes. Uh it's cold out here, and uh, and I went to try and get a taxi, they were all running away from me.". Being treated poorly made participants feel abandoned by society, and participants who experienced negative treatment grew to distrust service providers. It did not help that in some shelters service providers infantilized them, with participants describing how they would wake up in the middle of the night and come downstairs only to be berated and told to return to bed like a child. Husky described how this broke down trust, saying about service providers, "…some of us have a hard time accepting the fact that you are there, there, they're there to help us…Because there's also a trust barrier." [Husky]. For participants who had grown to accept that the world was uncaring and unsafe, instances of support could not be easily trusted because they no longer fit into their understanding of the world.

Conversely, some participants described how when service providers treated them compassionately it enhanced their trust in the system, which was beneficial for developing ontological security. Casey Jones, describing the process of moving into transitional housing, discussed the importance of having a worker that she knew and trusted, saying, "She made me feel like I was home.". Interacting with a trusted service provider allowed Casey Jones to feel safer placing her trust in the system, which allowed her to begin to gain some stability. Tye talked about how his support worker was a trusted confidant and how "I tell him everything". While Tye described multiple instances of gaining and then loosing housing, he indicated he generally felt supported by the system due to his positive relationship with his support worker. Both Bambi and Lola described several instances of people treating them well, making the experience of homelessness more positive, and discussed how the shelter workers really looked out for them. For example, Lola, described how when she first lost her housing a security guard at the train station "…stayed with me until they [people from a community organization] came to pick me up because he didn't wanna leave me by myself.". Both Bambi and Lola indicated that while poor health had disrupted their sense of ontological security, their general good treatment from service providers allowed them to maintain some trust in social systems. Service providers, as representatives of systems, appeared to have influence over how participants saw those systems, and compassionate treatment could go a long way in enhancing ontological security.

**Theme 3: The importance of social connection in rebuilding ontological security.** Participants described in detail how moving into housing following homelessness was challenging to adapt to as routines and truths they had previously relied on were disrupted, and previous experiences of trauma made it difficult to trust and feel safe. Throughout this experience, participants were hindered or supported by their social connections. For some participants, building new social connections enabled them to rebuild a sense of ontological security after they found housing. For others, the lack of social connections and support hindered this process and resulted in loss of housing. Finally, a group of participants felt unable to transition to housing due to the social connections they developed while unhoused.

Participants discussed how moving into housing was mentally challenging but how, when housing was of good quality and support was provided, this transition could also provide an opportunity to establish a new sense of ontological security. This could be seen in the examples of Casey Jones, Gabriella, and Doc. Casey Jones was living in transitional housing at the time of the study and discussed how, while she appreciated how the people in the building tried to support one another, it was taking a long time to get used to. Having previous experiences of losing people and having people break her trust, Casey Jones stated "I've gotten to the point in life where I feel like why bother then. Like if I'm just gonna, cause I got too many disappointments". However, she indicated that living with supportive and kind people was making her question this decision. In transitional housing, Casey Jones was given ample time to build trust with new people and was supported through this process by staff in the building. Gabriella had a similar story. Gabriella, who at the time of the study moved into what she described as "really nice" housing stated, "I think it was just I was definitely feeling overwhelmed". Specifically, Gabriella was overwhelmed by establishing new routines and learning how to feel safe in a new environment. Feeling overwhelmed and uncertain in her new housing led to feelings of guilt, depression, and suicidal thoughts. Gabriella was supported through this transition by her best friend, cat, and support worker who, over time helped her feel comfortable in her housing. While at the time of the study Doc was living in poor-quality housing, his story also demonstrated the importance of social connection in rebuilding a sense of ontological security. Doc explained how he frequented a specific restaurant, despite the high cost of drinks there, because the servers recognized him, knew his order, and made him feel seen and welcomed. Doc described how being treated with dignity, and being seen as a whole person, enhanced his sense of self, which is important for developing ontological security.

Unlike Casey Jones and Gabreilla, Tye and Pekoe did not receive adequate support transitioning to housing. Tye discussed how it was hard to get used to having housing, and that it disrupted his routines and led to overusing substances, saying about his experience of moving to housing, "It was working a little too much, you know, because then I think I was starting to get back into maybe the drugs, or the alcohol and stuff.". Using substances was harmful for Tye's sense of self as he professed "I don't wanna become a druggie". Tye was not supported through these challenges and left his

housing in order to try and preserve his self identity, an important element of ontological security. Pekoe, who was renting poor quality housing at the time of the study and living with a roommate, described how living with someone who had also experienced trauma meant that "we're both…walking on eggshells". Pekoe indicated neither him nor his roommate were supported with their mental health and that led to him feeling constantly unsafe, thus prohibiting him from establishing a sense of ontological security. Feeling unsafe in his housing, and lacking support through this process, Pekoe was strongly considering leaving his housing. This story was echoed by the experience of another participant, Husky, who also described having left housing due to lack of support with trauma and a roommate who reminded him of his abusive father. While roommates with similar experience have the potential to be a source of support and understanding, the participants indicated that lack of mental health support and support in navigating challenges meant instead roommates exacerbated trauma and led to difficulty in developing ontological security once housed.

For participants who had been unhoused for most of their life, the idea of transitioning to housing was seen as a threat to their sense of ontological security. Three participants in particular, Donney, Smiley, and Husky discussed how to adapt to the reality of living unhoused, they had taken on roles within their community as protectors. Smiley indicated being a protector was important to his sense of self, saying "It's how I've always lived, it's how I was taught by my mother and grandmother and my grand, is to take care of others". Being a protector allowed Smiley to maintain a sense of identity which was beneficial for his sense of ontological security. Husky discussed how he needed to protect his community because of the social injustices they were all facing, saying "I am hearing the cries of us all". Husky had also had some prior bad experiences in housing and indicated he generally felt safer unhoused in familiar environments he knew how to navigate. The sentiment of putting the community first was also expressed by Donney, who said, "I can't even see myself moving into a place…We're one big family out here". Donney could not see himself moving away from his community as this would disrupt his sense of identity and the routines he had established. For Donney, Smily, and Husky their self-identity as protectors meant that securing a tenancy was not an option if it did not allow them to support their community.

## Discussion

The purpose of this study was to explore how older adults experienced trauma across the transition to housing following homelessness from the perspective of ontological security. The lens of ontological security contributes to understanding older adults across the process of finding and transitioning to housing, and how systemic factors are experienced as traumatic throughout this experience.

Policy makers frequently describe some individuals who experience chronic homelessness as 'housing resistant' [62,63], ignoring the many barriers to finding stable housing [64]. A key barrier that is well documented in the literature is poor-quality housing [58,65–67]. In the current study poor quality housing intersected with aging to add the consideration of accessible housing. For example, participants described being shown housing with stairs, despite having difficulty with mobility and many seemed to have given up on trying to find housing that would accommodate devices such as grab bars or raised toilet seats despite clearly needing them. Further, health concerns were sometimes distracting, leaving participants little time and energy to focus on finding housing. For those who were actively searching for housing, instead of focusing on what they really needed, participants were forced to choose between forms of inadequate housing options, such as housing with bed bugs or unsafe roommates. Through the perspective of ontological security [29,30], it also became apparent that participants were trying to make choices that best maintained their sense of self. For example, some participants turned down poor-quality housing in an effort to maintain a sense of self-respect, which can be particularly challenging to maintain as an older adult in a youth-centric society [68]. In other cases, participants had learned to adapt to being unhoused by self-identifying as community protectors. Hawkins et al., [34] describe three important aspects of building a sense of ontological security: establishing a new sense of identity, community, and safety. Through identifying as community protectors these participants engaged in all three, creating community, a sense of identity, and safety in numbers. Building a sense of ontological security supported participants in living unhoused, however, it also hindered their

ability to transition to housing. The findings of this study therefore add to the discussion around barriers to housing and indicate the need for housing that is accessible, good quality, supports participants' sense of identity, and enables continued community.

Authors have argued that across the transition to housing following homelessness, individuals need mental health supports, opportunities for meaningful activities and work, stability and control, safety, and opportunities for connection and belonging [66,67,69]. The findings from this study support these arguments and add, from the perspective of ontological security, the need to support older adults through the transitional period of finding new housing to foster a new sense of ontological security. Some participants managed to find good quality and accessible housing, however, adapting to housing was challenging. Seen through the lens of ontological security [29,30] such a finding is unsurprising. Significant life transitions, including moving can be challenging at the best of times, and moving to a safe place after living in an unsafe environment can be difficult to get used to [11,70,71]. In this study the change associated with finding housing disrupted routines, leading to increased substance use, distress, and suicidal thoughts. The transition to good quality housing was therefore not so simple as moving into housing but involved a lengthy process of learning to adapt to a new reality. Participants need stable housing with social supports, opportunities to build control [67] and understanding during the process of adapting. Service providers and individuals transitioning to housing following homelessness might benefit from recognizing that building a new sense of ontological security is a process and that it might take a while to feel comfortable within a new environment.

### Research recommendations

Findings from the current study indicate the need for continued research exploring how older adults experience trauma across the transition to housing following homelessness. Specifically, research needs to focus on three key areas. Firstly, Giddens' [29,30] theory of ontological security was a good match for the findings and extended our understanding of establishing security. Future researchers could explore facilitators and barriers to establishing a new sense of ontological security across the transition to housing following homelessness. Secondly, findings from this study indicate the need to evaluate interventions for older adults transitioning to housing following homelessness. A recent study in Australia took a quantitative approach to explore a long-term care home specifically designed for those with previous experiences of homelessness [72], however it is also important to look at participant experiences of such interventions from a trauma-and-violence informed perspective. Taking a trauma and violence informed perspective is important because, similar to previous studies [15,66], the findings of the current study indicate that following the transition to housing, participants experienced increased substance use, suicidal thoughts, guilt, and re-emergence of trauma memories. Finally, consistent with the model proposed by Humphries et al., [73], findings from this study indicate that older adults who experienced homelessness prior to age 50 were not receiving the support they needed to stay housed. Research therefore needs to focus specifically on the needs of older adults who have experienced chronic/episodic homelessness across the life course separately from older adults experiencing their first episode of homelessness.

### Practice recommendations

The current study highlights the importance of implementing trauma and violence informed practices [17] when working with older adults with current or previous experience of homelessness. When discussing trauma-informed care, Hopper et al., [9] indicates the importance of providing safe and supportive environments across the experience of homelessness that allow individuals to rebuild control, maintain boundaries, and utilize their skills. Elaborating on trauma informed care, Wathen and Varcoe [17] have further contended that service providers need to understand the context of structural violence in which people experiencing homelessness live. The findings of the current study indicate that these suggestions are relevant to older adults.

Homelessness is a form of structural violence that many of the older adults in this study experienced as traumatic, both for themselves and for their community. Unhoused individuals experience stigma and discrimination from most of society [60] and for older adults who have lived unhoused for long periods of time these experiences can lead to what some participants described as a trust barrier. For older adults who were new to living unhoused, their trust in the system was quickly decreased by being offered poor quality inaccessible housing and being constantly rejected by potential landlords. Loss of trust in a system they had thought would be supportive was psychologically difficult to handle and led participants to express feeling worthless, and as if they were stuck in a hopeless situation. Service providers who interact with older adults with experiences of homelessness therefore might consider that trauma-and violence informed care is not only about recognizing adverse childhood experiences, but also the everyday trauma and structural violence of living unhoused and in poor-quality housing.

Moving into good quality housing is a huge life transition that can lead some older adults to increased substance use or suicidal thoughts and feelings of guilt. When a person has been treated poorly for long periods of time, as was the case for many of the older adults in this study, simply securing a tenancy was not enough to feel secure and safe, and instead led some to feel unmoored and uncertain about how to live in their new reality. However, participants who successfully transitioned to housing indicated that social support from service providers and the community were instrumental in this success. Service providers can therefor have a huge impact on older adults' ability to adjust to housing. Prior research indicates service providers can support individuals through providing mental health supports, opportunities for meaningful activities and work, stability and control, safety, and opportunities for connection and belonging [66,67,69]. The findings of this study support those recommendations and additionally indicate participants needed support in finding accessible housing, and education and community to normalize the challenges in adjusting to housing. Further, some individuals who had been homeless for a long time expressed how they viewed themselves as protectors for their communities, but felt that service providers hindered, rather than supported them in their role as protectors through service restrictions and distrust. While this is an area that require further study, these preliminary findings indicate a need for novel strategies to support these individuals such as supporting communities to move to housing rather than individuals.

## Policy recommendations

There is a dearth of good quality accessible housing that is deeply affordable, and thus available for individuals living in poverty [58,59]. Policy makers should recognize that without deeply affordable, good quality accessible housing with mental health and physical support, older adults will continue to be forced to live in inhumane living conditions and struggle to thrive following homelessness [13,58,59]. While struggling to thrive following homelessness is unfortunately common for individuals transitioning to housing [13], older adults have unique needs that warrant further attention. Specifically, some older adults expressed a desire to move to supportive housing and others needed support to age in the right place following the transition to housing.

Some older adults in this study indicated a desire to move into housing that more fully supported their physical and mental health needs. Specifically, one participant discussed at length her desire to move back into long-term care. However, amid recent reports indicating that long-term care in Canada is rife with abuse [74], this feels ill-advised. Instead, permanent supportive housing shows promise [75]. Importantly, when designing permanent supportive housing, policy makers need to ensure such programs are aligned with the principles of trauma and violence informed care [17]. This includes collaboration with older adults with experiences of homelessness during the program design process and ensuring all staff are trained in trauma and violence informed practices.

Another avenue through which to help older adults transition out of homelessness is through putting resources into supporting older adults to age in the right place. Aging in the right place refers to the ability of older adults to age in the community [76], and the experience of older adults transitioning to housing in this study illuminate some specific aspects that are needed for this to occur. Firstly, resources need to be put into building more deeply affordable and accessible

housing [77,78]. Specifically, older adults indicated that they needed housing that was wheelchair or walker accessible with no stairs, had grab bars in the bathrooms, and was near resources such as grocery stores. Others have discussed these forms of accessible housing as having a universal design [79,80] suited to the needs of older adults. Within this study, older adults also discussed needs for support including mental health support, and support within the home such as cleaning, cooking, and shopping which older adults described as increasingly challenging due to changes in mobility. Policy makers need to recognize that older adults transitioning to housing following homelessness require more support than is currently available for people on a low-income, and older adults transitioning to housing following homelessness need to be connected with services to allow them to live with dignity.

## Limitations

While there are many reasons to conduct a secondary analysis, there are also some limitations [44]. Because we were unable to pose questions to participants directly relating to trauma, we had a limited ability to explore this construct directly. Another limitation was the lack of diversity within the sample. The majority of the older adults who participated in this study were white and did not identify as 2SLGBTQIA+. As such, when considering the transferability of these findings, readers should consider that racialized older adults and older adults who identify as 2SLGBTQIA+ might have different experiences. Finally, while the analysis indicated possible differences between people who aged in homelessness and those who become homeless for the first time as older adults, we were not able to explore this difference.

## Conclusion

Older adults within this study experienced trauma across the transition to housing in a myriad of ways, all underpinned by the loss and subsequent struggle to regain a sense of ontological security. Experiences of aging, lifetime traumatic experiences, and structural violence all were causes of the loss of ontological security. The transition to housing represented an opportunity to rebuild a sense of ontological security, however only when housing was affordable, accessible, and safe, and participants were emotionally supported. The experiences of trauma discussed by the participants indicates the need for continued use of trauma-and-violence informed care [17] on the part of service providers. Participants indicated that the experience of homelessness and transitioning to housing was fraught with trauma, including the stigma and discrimination felt from service providers and the broader community. When service providers were caring and sincere however, participants indicated that it made a huge difference. Finally, policy makers need to consider that older adults have different needs than the general population of individuals experiencing homelessness. Older adults described needing accessible housing and support with activities of daily living. Policy makers need to redouble efforts to ensure low-income older adults are provided with the necessities to age-in the right place [76] as both a preventative measure and across the transition to housing following homelessness.

## Supporting information

**S1 Text. Appendix A which contains the interview protocol.**
(DOCX)

## Author contributions

**Conceptualization:** Rebecca Goldszmidt, Carrie Anne Marshall.

**Data curation:** Carrie Anne Marshall.

**Formal analysis:** Rebecca Goldszmidt.

**Investigation:** Rebecca Goldszmidt.

**Methodology:** Rebecca Goldszmidt.

**Project administration:** Rebecca Goldszmidt.

**Resources:** Rebecca Goldszmidt.

**Supervision:** Carrie Anne Marshall.

**Validation:** Rebecca Goldszmidt.

**Visualization:** Rebecca Goldszmidt.

**Writing – original draft:** Rebecca Goldszmidt.

**Writing – review & editing:** Shu-Ping Chen, Rebecca Gewurtz, Carri Hand, Carrie Anne Marshall.

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
