## [Decision Letter · Decision Letter 0]

13 Jun 2025

PMEN-D-25-00136

Older Adults Experiences of Transitioning to Housing Following Homelessness from a Perspective of Ontological Security: A Secondary Analysis

PLOS Mental Health

Dear Dr. Goldszmidt,

Thank you for submitting your manuscript to PLOS Mental Health and I am sorry for the delay in reaching a decision. Thank you for your patience. After careful consideration of the reviewer reports, we feel that your paper has merit but does not yet fully meet PLOS Mental Health’s publication criteria as it currently stands. Therefore, we invite you to submit a revised version of the manuscript that addresses the points raised during the review process.

Please address all of the comments raised by the reviewers, which you can find at the end of this email and in the attachment.

We look forward to receiving your revised manuscript.

Kind regards,

Karli Montague-Cardoso

Executive Editor

PLOS Mental Health

Journal Requirements:

Additional Editor Comments (if provided):

Reviewers' comments:

Reviewer's Responses to Questions

**Comments to the Author**

1. Does this manuscript meet PLOS Mental Health’s publication criteria?

Reviewer #1: Yes

Reviewer #2: Partly

2. Has the statistical analysis been performed appropriately and rigorously?

Reviewer #1: N/A

Reviewer #2: Yes

3. Have the authors made all data underlying the findings in their manuscript fully available (please refer to the Data Availability Statement at the start of the manuscript PDF file)?

Reviewer #1: Yes

Reviewer #2: No

4. Is the manuscript presented in an intelligible fashion and written in standard English?

Reviewer #1: Yes

Reviewer #2: Yes

Reviewer #1: Thank you for your submission related to using ontological security to understand older adults transitions out of homelessness. It was an interesting read, well structured and well aligned to the journal you have submitted to. There are some changes required before I can recommend this for publication, however, this is overall a well thought-out article which highlights an under-researched area of homelessness research. There are a number of suggested changes below, however, I feel this would still class as 'minor' revisions as the underlying justification and argument made in the article is well done. I note that the lead author is an early career researcher and would like to recognise the effort that I have no doubt went into writing this article.

Abstract: ‘deeply affordable’ - rephrase for clarity.

Introduction/context: Trauma can be understood differently - it would be good to acknowledge differing understandings of trauma and how this concept has been applied for the purpose of this article.

Some discussion on housing systems and homelessness in study areas would be beneficial to help set the scene.

Methods: I’d like more information on how participants were recruited in the primary study - how were they accessed, use of gatekeepers, etc.

States trauma could be seen as an underlying presence throughout all interviews - in what way?

Findings:

Justify phrasing - ‘when entire world is upended’ in relation to homelessness and transitions with reference to housing literature and ontological security in intro section to qualitative findings

Is it reasonable to say ‘losing a sense of ontological security’ as this suggests it was there, and then it wasn’t. Ontological security can be seen as more of a continuum or process rather than something that is simply lost or found. Pg. 13

‘Focus was spent worrying about his health’ - phrasing. Pg. 13

Smiley’s experience on Pg. 14 could be developed for clarity - unclear why not allowed to stay - had they previously been banned?

More could be said about how derelict housing options impact ontological security to better develop argument. Pg. 15

Point to develop - social structures designed to constrain and prohibit rather than support - to add how removal of possessions impacts feelings of security and being in control of one’s own life?

Pg. 19 - ‘faced with the uncertainty of their lives’ - clarity - do you mean lack of control over their lives? Or do you mean they life/health? I think this can be better phrased to make this distinction clear.

‘When the world felt like it was falling apart’ - is this something a participant said or something assumed? Can this be made clear or rephrased? As you have stated, some people are able to build feelings of ontological security in places that could be seen as inherently insecure. This suggests resilience to hardship rather than the world falling apart.

There was a general lack of citations in the results section that could have supported some of the discussion/arguments made.

Recommendations could be more carefully phrased - overly emotive language in places that is not always supported by the discussion above. Some of the recommendations are not discussed in the results or discussion - e.g. wish to move to supported accommodation. Rather than introduce new ideas in the recommendations, these should be in the discussion and results too or not part of the recommendations.

Overall, minor points: more care to be taken in phrasing throughout - some terminology used is more emotive than balanced. There were some minor spelling/punctuation mistakes which can be resolved in proofreading.

Reviewer #2: PLOS Mental Health Review – Older Adults, Transitions out of Homelessness, and Ontological Security

Introduction:

I like that you connect trauma to ontological security. But to that point, in your introduction, I would keep that as your focus. Instead of saying that OS has been applied to ageing, trauma, and housing insecurity reframe this as a few examples of trauma and OS being connected… ex: when someone has a stroke, when someone lives though a natural disaster, when someone is facing long term housing insecurity. These all seem to be instances of trauma.

Literature Review:

I am not sure why there is no literature review.

I think you could have a very short one that goes into detail about housing insecurity as trauma that then affects OS.

You could have the experiences of homelessness in here. I know there is a lot of literature on homelessness, trauma, and OS. Maybe look at Kim Skobba to help you start? And then go into the specifics you mention in your intro, that people have looked at the trauma older adults particularly experience while unhoused

And then you could talk about how transitioning to housing can be traumatizing for people exiting homelessness ending with the gap you mention in your introduction, that there is not enough literature on how this specifically impacts the growing number of older adults experiencing homelessness

One suggestion I have is to look at any work that exists on older adults living in PSH or SROs (I know there is literature on these types of housing options for older adults and how they feel within them – but there will not be a lot to your point of this being understudied at a moment when we are going to see a lot more senior homelessness…)

Methodology:

I would say more about secondary analysis. Maybe define it. And say why it is a good method given your circumstances. Totally get it if you were a student RA but I would just spell it out a bit more. SA is… and it was ideal for me because… in fact it gets used a lot as a method because… Also in your introductory paragraph it might be helpful for you to briefly describe the project that the data came from… just so we can follow along as your readers… why this secondary analysis of this data helped you close the gap in OS research you mention. I am assuming it was a study on older adults transitioning to housing from homelessness but spelling it out helps readers follow along seamlessly

I really appreciate your positionality section. I found a small typo for you! “There are also commonalities that led to us…”

I would also consider putting your sample n in the methods section… I was looking for it and then found it in the findings section.

Findings:

I like seeing your sample demographic characteristics.

Now that I see 10 of your 15 interviewees were unhoused and 5 were housed, I would reframe your intro… “not enough has been written on how older adults experience the trauma of homelessness or that transition to housing…” Right now with it as “the transition to housing” only I assumed your 15 were all now housed…

At the top of qualitative findings, I think you mean to say (first sentence under Essence) “the essence of the experience of transitioning from housed to unhoused… and then back to housing…” Or maybe you want to start with “going from housed to unhoused” means you learn to live in a new reality… and this is true to even as people rehouse…

When you go into ageing and physical distress I would connect this more clearly to being unhoused… this sounds like something we all may go through when we age… feeling a loss of OS as we worry more about our health, but this is compounded when you are unhoused… and then of course you rightfully mention your preoccupation with health can take up all your time and make it hard to find housing…

Did Lola find housing? It is not clear.

I think the one thing missing from Bambi’s story is if the shelter makes her feel less OS… this would keep your through line of homelessness and trauma

I really like the distinction you make between people who have been unhoused for a while and have “aged on the street” so to speak versus those who have just become unhoused as seniors… both are growing trends… and both have their own traumas…

When you get to the part about systems being broken and specifically people feeling that they weren’t having their needs met in health settings, I would more solidly connect this to the experience of homelessness… why is it that unhoused individuals are having these poor interactions with psychiatrists?

Why did Gabriella feel overwhelmed in her new housing?

In your learning to adapt to a new reality section it currently feels like you are doing too many things… 1) you are talking about how some people could build OS in new housing settings and how others could not on account of things like the presence of social support… but then you also discuss how people on the streets built OS together through community and are now loth to enter housing… but then you also go into what people do when they can’t improve their OS in terms of coping mechanisms… I would reorganize this section… What is it that your interviews show you? Some people redevelop OS when they move into housing because there are social supports? Others don’t because there are no social supports? By contrast those who are not connecting to housing at least have each other (i.e. social supports/community) and that gives them a new sense of OS even if it makes it harder for them to move into housing (because they don’t want to leave their friends behind? Which I have seen!)… this all to me demonstrates points of action we could take… all of which revolve around social ties…we could build up social support in PSH… move entire encampments into PSH together… maybe you want to rework this last section of your findings to focus on the through line… the importance of social supports for rebuilding OS… and maybe wait on the other stuff about acceptance and put it in your discussion…

Discussion:

I think you need to go back through your findings section and discussion section together and really think about what it is you found and are thus arguing. It is just a little disjointed at the moment. And when you do this I think you also need to keep the thread of “older adults” in mind… why is this specifically important for them (declining health and ability… already traumatic possibly but worse when unhoused… declining social networks with people in their lives dying/divorces…this then makes your findings more interesting for older adults… ex: why social supports may be so important for rebuilding OS either in housing or on the streets…ex: why having the rug pulled out from under you is especially scary at this age, when you have health conditions and lost abilities… )

**Do you want your identity to be public for this peer review?** For information about this choice, including consent withdrawal, please see our Privacy Policy

Reviewer #1: **Yes: ** Dr. Kirsty Cameron

Reviewer #2: No

---

## [Decision Letter · Decision Letter 1]

14 Aug 2025

PMEN-D-25-00136R1

Older Adults Experiences of Transitioning to Housing Following Homelessness from a Perspective of Ontological Security: A Secondary Analysis

PLOS Mental Health

Dear Dr. Goldszmidt,

Thank you for submitting your revised manuscript to PLOS Mental Health. After careful consideration by the reviewer and our editorial team, we would like to offer you one final round of minor revisions in order to give you the opportunity to incorporate the remaining comments from the reviewer, which you can find below. I will then assess the changes in-house upon resubmission as opposed to sending this back to review. If you have any questions at all, please feel free to reach out to kmontague-cardoso@plos.org and I will be happy to assist you.

We look forward to receiving your revised manuscript.

Kind regards,

Dr Karli Montague-Cardoso

Executive Editor

PLOS Mental Health

Journal Requirements:

Additional Editor Comments (if provided):

Reviewers' comments:

Reviewer's Responses to Questions

**Comments to the Author**

Reviewer #2: (No Response)

publication criteria?

Reviewer #2: Yes

3. Has the statistical analysis been performed appropriately and rigorously?

Reviewer #2: Yes

4. Have the authors made all data underlying the findings in their manuscript fully available (please refer to the Data Availability Statement at the start of the manuscript PDF file)?

Reviewer #2: No

5. Is the manuscript presented in an intelligible fashion and written in standard English?

Reviewer #2: Yes

Reviewer #2: Introduction and Lit Review

I really like your reworking of your introduction. I enjoy the way you frame trauma as structural in origin but deeply personal in how it is experienced, and how you lay out the myriad consequences of trauma for individuals. This lays the foundation for your topic - homelessness as structural trauma but also housing as a mediating factor... poor housing options are still structural trauma... thank you for also taking your range of topics related to ontological security (aging, homelessness, etc.) and just referring to how they all can produce trauma. I think it streamlines your paper. Thank you also for adding in a bit more about how homelessness affects OS (Skobba etc).

My one suggestion here is to divide up your intro and lit sections. They are now one 4 page section. Having a separate intro that lays out your framework concisely and a roadmap for the rest of your paper could help readers better follow your arguments. And then you could have a separate lit review where you dive deeper into how trauma is structural in origin, how it affects individuals in multiple ways, why it is related to ontological security... and then how trauma and ontological security affect older adults experiencing homelessness.

If you find this feedback useful, you could also have a very abridged definition of ontological security in your intro so people know right away what it is - you can use this one you wrote - Ontological security is the sense that the world and the way that it is constructed in one’s personal life can be depended upon to remain constant (29,30). And then you can expand on what it means (the stuff you write about Giddens for example) in the lit review. Just a thought. I see though I made the same comment last time - so maybe this is a style discrepancy we have across our disciplines.

One final thing: the way you so clearly define trauma now as structural in origin it still lends itself to you thinking about aging, natural disasters, and housing insecurity as instances of trauma that affect OS but maybe you want to be clear that systems of racism, classism, etc. are what put certain people at greater risk of natural disaster or having a stroke.

Methods

Thank you for defining secondary analysis and giving context - benefits of SA, challenges of SA, and why you could evade those challenges given your particular circumstances.

Thank you for clarifying what the original study was. It just helps readers follow along in terms of why SA was such a good fit here.

Thank you for making your sampling strategy for SA easier to find by moving it to methods.

The last two sentences of your methods section are a little hard to read. Consider revising for clarity.

Following the guidance of (54), a central essence regarding how older adults

experience trauma cross the transition to housing following homelessness was created that

connected all the themes. The final stage of the analysis was the writing, where we described

what each theme is about, using participant quotes to center the participants’ experiences.

Findings

Thank you for now being clear in your intro section and elsewhere that these are folks transitioning from homelessness to housing. This way it encompasses all your interviewees, including those still unhoused.

I can't remember from the first review, but this works: mentioning your sample came from an original sample and that it is n=15 in the methods section but then going into demographic info for these 15 in the findings.

How does Lola's focus on living life in the moment make it hard for her to find housing?

Reviewing your findings again, I think you want to take another look at your framework... I am getting this picture as I read: homelessness and even the transition out of homelessness impacts OS as homelessness is traumatizing and so too is looking over subpar housing options where you feel unsafe... and what is more, when you are homeless and dealing with health issues or childhood trauma that led to homelessness these things not only erode OS but make it hard to get into housing. What helps? When people fight back against violent structures, when people have social support, and when people have adequate housing options available to them. I don't think it is wrong to have all this - I would just have you review your framework. Right now your research question is: How do older adults experience trauma across the transition to housing following homelessness? But maybe it is "how are older adults affected by trauma and what protective factors exist for them?"

Maybe I am thinking too deeply about this, but in your theme 2, "the system is not going to save me," this comes across for me in two ways 1) the system isn't ending my homelessness and 2) the system - ie multiple systems that interact with the unhoused, healthcare and homelessness services here specifically, is not just not ending homelessness but furthering suffering through stigmatization. It's important that your interviewees mention stigma at hospitals and at shelters. And then you end on a note where stigma is not happening, where people are treated with dignity which is so important for OS (a sense of stability in your life, a sense of stability in your identity - that these people don't see me as homeless but as a whole person). But I think maybe because you are focused on "the system" here you might want to nix the example of servers treating someone nice and focus on moments in "the system" where providers are compassionate. And on this note, maybe it would help readers follow your argument across all three themes if you clearly connect your findings to OS... like the above example I gave (when you are treated with dignity, you can maintain your sense of self...). Or how theme 1 "having the rug come out from other you" throws your sense of constancy in your life/constancy in your self out the window...

For your third theme, I have a similar comment. I would take the time to analyze each example you share. Ex: the two roommates who are both traumatized... why is it that this social tie actually works against OS... this is not a lot of work, just a sentence after each example. Does this person feel unsafe because of their roommate? Explain! It just is that much effort. Like a sentence.

And I love the finding you have that for some unsheltered individuals they build new OS together on the streets. But this makes it hard to move into housing. I think you pared that section down a bit too much. I would put that explanation of your examples back in.

I also think it was smart to take out the coping part in theme 3. Your through line is the social ties - whether they are present or not/helpful or not for maintaining housing and how they are protective on the streets but prevent housing.

SUMMARY OF RECOMMENDATIONS

In conclusion, I don't know if you need to make a lit review. This might be a style discrepancy we are butting up against. But I do recommend putting in a line about how aging, natural disasters, and housing are all examples of structural trauma. Feel free to keep or exclude my recommendation to rework your last two sentences in Methods. I think the through line in my comments about your Findings section is just to add a bit of explanation after your examples. Be explicit. Why is this example an example of eroding OS or an example of it being preserved? I would keep this really short. Just a comb through and adding the sentences of explanation where you need them. I really hope this helps!

**Do you want your identity to be public for this peer review?** For information about this choice, including consent withdrawal, please see our Privacy Policy

Reviewer #2: No

---

## [Decision Letter · Decision Letter 2]

10 Sep 2025

Older Adults Experiences of Transitioning to Housing Following Homelessness from a Perspective of Ontological Security: A Secondary Analysis

PMEN-D-25-00136R2

Dear Ms Goldszmidt,

We are pleased to inform you that your manuscript 'Older Adults Experiences of Transitioning to Housing Following Homelessness from a Perspective of Ontological Security: A Secondary Analysis' has been provisionally accepted for publication in PLOS Mental Health.

Best regards,

Karli Montague-Cardoso

Staff Editor

PLOS Mental Health

Reviewer #2:

Reviewer Comments (if any, and for reference):

Reviewer's Responses to Questions

**Comments to the Author**

Reviewer #2: (No Response)

publication criteria?

Reviewer #2: Yes

3. Has the statistical analysis been performed appropriately and rigorously?

Reviewer #2: Yes

4. Have the authors made all data underlying the findings in their manuscript fully available (please refer to the Data Availability Statement at the start of the manuscript PDF file)?

Reviewer #2: No

5. Is the manuscript presented in an intelligible fashion and written in standard English?

Reviewer #2: Yes

Reviewer #2: Quick Style Tips/Editing

At first I thought it was a style choice, but now I see sometimes you have a period at the end of a quote and then another period after the quotation marks and sometimes you do not. I would just go in and pick one style.

Losing not loosing on top of page 22.

Quick Substantive Feedback

I really like your article. I would just recommend that, in your Discussion section, you now tie it all together. In your Intro/Lit section you define OS as a sense of constancy in one's self-identity and material and social environments (Giddens). It is clear in your findings how self-identity and material and social environments were destabilized and sometimes restabilized. I don't think you need to go through and be explicit throughout your findings, but I would be explicit in your Discussion. Your Discussion section is a chance to connect your Intro/Lit section/framework for your paper to the findings you share. You are explicit in your discussion about self-identity being destabilized by homelessness, but maybe you want to do it with material and social environments as well, especially because it is really easy to go in and add a bit on that. People are definitely not going to want poor housing when it erodes their sense of self (i.e. I deserve better than this). So yes, you can argue they are preserving their sense of OS by turning that down. But for those who take that substandard housing, they don't feel secure in their material environment. You mention this in your findings a lot, but you can maybe go into your Discussion section and really connect all of your findings to your original framework of OS/Gidden's definition. Similarly, poor treatment at the hands of providers because of stigma means you don't feel a sense of constancy in your social environment, which as you rightfully say, makes it hard for you to hold onto your sene of self. It is just a matter of going back over your Discussion section and making sure each piece of Giddens definition you are using is being connected to your findings (it is there in your findings for sure, just make it really clear in your Discussion). On this same note, some folks who are chronically homeless have that sense of constancy in their social environment/social relationships, it's just on the streets...you talk about their self-identities as protectors giving them a sense of OS and I think that is great, I just think you could very easily go in and use Giddens full definition in your Discussion to tie it all together. People experiencing chronic homelessness sometimes have a sense of OS because of a sense of self-identity and really supportive social ties. And I totally agree that does not get looked at enough as a barrier to housing (and people have suggested moving whole encampments together inside)! They are doing it where I'm at...

Other than that, in your OS and Homelessness section you start with an official definition of homelessness that really is just about housing. Then you say "as is clear in the definition, homelessness is not just about housing (it's about a loss of identity/social stigma... getting more into OS as defined by Giddens - not just loss of material constancy of one's housing, but of social constancy as you are now stigmatized and maybe lose family and friends and of your self-identity). This is a quick fix. I think you just mean, "But this is not the whole story..." rather than "As is clear in this definition..."

If you make these changes, I would say this paper is ready for publication.

**Do you want your identity to be public for this peer review?** For information about this choice, including consent withdrawal, please see our Privacy Policy

Reviewer #2: No
